# Internalization of *Clostridium botulinum* C2 Toxin Is Regulated by Cathepsin B Released from Lysosomes

**DOI:** 10.3390/toxins13040272

**Published:** 2021-04-09

**Authors:** Masahiro Nagahama, Keiko Kobayashi, Sadayuki Ochi, Masaya Takehara

**Affiliations:** 1Department of Microbiology, Faculty of Pharmaceutical Sciences, Tokushima Bunri University, Yamashiro-cho, Tokushima 770-8514, Japan; kobakei@ph.bunri-u.ac.jp (K.K.); mtakehara@ph.bunri-u.ac.jp (M.T.); 2Faculty of Pharmacy, Yokohama University of Pharmacy, 601 Matano-cho, Totsuka-ku, Yokohama-shi, Kanagawa 245-0066, Japan; sadayuki.ochi@hamayaku.ac.jp

**Keywords:** *C. botulinum* C2 toxin, internalization, cathepsin B

## Abstract

*Clostridium botulinum* C2 toxin is a clostridial binary toxin consisting of actin ADP-ribosyltransferase (C2I) and C2II binding components. Activated C2II (C2IIa) binds to cellular receptors and forms oligomer in membrane rafts. C2IIa oligomer assembles with C2I and contributes to the transport of C2I into the cytoplasm of host cells. C2IIa induces Ca^2+^-induced lysosomal exocytosis, extracellular release of the acid sphingomyelinase (ASMase), and membrane invagination and endocytosis through generating ceramides in the membrane by ASMase. Here, we reveal that C2 toxin requires the lysosomal enzyme cathepsin B (CTSB) during endocytosis. Lysosomes are a rich source of proteases, containing cysteine protease CTSB and cathepsin L (CTSL), and aspartyl protease cathepsin D (CTSD). Cysteine protease inhibitor E64 blocked C2 toxin-induced cell rounding, but aspartyl protease inhibitor pepstatin-A did not. E64 inhibited the C2IIa-promoted extracellular ASMase activity, indicating that the protease contributes to the activation of ASMase. C2IIa induced the extracellular release of CTSB and CTSL, but not CTSD. CTSB knockdown by siRNA suppressed C2 toxin-caused cytotoxicity, but not siCTSL. These findings demonstrate that CTSB is important for effective cellular entry of C2 toxin into cells through increasing ASMase activity.

## 1. Introduction

*Clostridium botulinum* C2 toxin is a member of the family of clostridial binary actin-ADP-ribosylating toxins. C2 toxin comprises an enzyme component (C2I) and a separated binding/translocation component (C2II) [1,2,3,4]. Internalization of C2I to host cells is performed with the proteolytically cleaved form of C2II (C2IIa). C2I catalyzes mono-ADP-ribosylation of actin monomer (G-actin) at Arg177, leading to filamentous actin (F-actin) depolymerization and cell morphological change (rounding) [1,3]. Other members of this toxin family are *Clostridium perfringens* iota-toxin, *Clostridium spiroforme* toxin, and *Clostridium difficile* ADP-ribosyltransferase [1,2,3,4].

C2IIa triggers the internalization of CI [1,2,3,4]. C2IIa recognizes an *N*-glycan containing the α-D-mannoside -β1, 2-*N*-acetylglucosamine motif, the cellular receptors, on target cell membranes, forms heptameric structures, and associates with C2I [5,6]. After endocytosis, the C2I/C2IIa complex is delivered to early and recycling endosomes through the host endosomal trafficking pathway [3,6,7]. C2IIa oligomers insert into the early endosomal membrane in acidic conditions and form pores, permitting the entry of C2I from endosomes to cytoplasm [1,2,3]. After translocation, C2I catalyzes ADP-ribosylation of G-actin in the cytosol [1,2,3]. A small amount of C2I and C2IIa is transported together back to the cell membranes by recycling endosomes, whereas other components are delivered to late endosomes/lysosomes for degradation [7].

Our previous reports have indicated that ASMase is needed for C2 toxin internalization. We have reported that C2IIa forms a heptamer on the MDCK cell membrane and induces the influx of Ca^2+^ into the cytosol [8]. C2IIa-induced Ca^2+^ influx into the cytosol triggers exocytosis of ASMase from lysosomes [8]. ASMase then hydolyzes sphingomyelin to ceramide on the outer leaflet of the cell membrane [9,10,11]. Ceramide-enriched membrane microdomains invaginate cytoplasm and facilitate C2 toxin endocytosis [8]. The lysosomes include more than fifty hydrolases [12]. It has been reported that the cathepsins exported from lysosomes are important for pore-forming toxin (PFT)-induced plasma membrane wound repair [13].

It is unclear whether extracellularly released cathepsins from lysosomes play a role in C2 toxin internalization in host cells. Epithelial Madin–Darby canine kidney (MDCK) cells are a useful tool for analyzing the entry of C2 toxin into cells [6,7,8]. In this study, we investigated whether the C2 toxin takes advantage of lysosomal proteases to promote this internalization into host cells.

## 2. Results

### 2.1. C2 Toxin-Induced Lysosomal Exocytosis

We previously reported that C2 toxin causes Ca^2+^ uptake, facilitating lysosomal exocytosis [8]. To evaluate whether C2IIa induces the lysosomal exocytosis, the liberation of lysosomal marker enzyme β-hexosaminidase (βHex) in culture medium of C2IIa-treated MDCK cells was evaluated (Figure 1A). When cells were treated with C2IIa, βHex activity in culture medium increased in a time-dependent manner. In contrast, C2IIa-induced βHex release decreased in the presence of lysosomal exocytosis inhibitor bromoenol lactone (BEL), indicating that lysosomal exocytosis is promoted by C2IIa. It has been reported that lysosomes are the origin of the proteases that modulate plasma membrane repair [13]. To investigate the role of lysosomal proteases, we assessed the activities of the predominant lysosomal proteases, i.e., cathepsin D, the main lysosomal aspartyl protease, and cathepsins B and L, the main lysosomal cysteine proteases [13]. After incubating MDCK cells with C2IIa, protease activity was evaluated in the culture medium for various periods, utilizing specific fluorescent substrates. As shown in Figure 1B, C2IIa induced a release of cathepsins B and L in a time-dependent manner, but not a release of cathepsin D. To investigate whether the cathepsins released by C2IIa are derived from lysosomes, we examined the C2IIa-induced cathepsin release in the presence of BEL (Figure 1C). Release of cathepsins B and L by C2IIa was inhibited by BEL, suggesting that C2IIa-triggered lysosomal exocytosis results in the release of cathepsins.

### 2.2. Lysosomal Proteases Are Needed for Internalization of C2 Toxin

To evaluate whether lysosomal proteases are responsible for the C2 toxin internalization, we studied the effects of serine, cysteine, and aspartic protease inhibitors on the cytotoxicity of C2 toxin. Aspartic protease inhibitor, pepstatin A (PepA), and serine protease inhibitor, 4-(2-aminoethyl)benzenesulfonylfluoride HCl (AEBSF), cannot affect C2 toxin activity (Figure 2A). In contrast, treatment of MDCK cells with cysteine protease inhibitor E64 led to decreased cytotoxicity of C2 toxin (Figure 2A). BEL also inhibited toxin-induced cytotoxicity (Figure 2A). Then, we investigated the effect of E64 on C2IIa internalization. In the cells preincubated with DMSO (Vehicle), C2IIa was found in cytoplasmic vesicles (Figure 2B). In contrast, in MDCK cells preincubated with E64, the cellular uptake of C2IIa was reduced, and C2IIa was present in the cytoplasmic membrane. These data show that lysosomal cysteine proteases are needed for C2 toxin endocytosis.

Lysosomal ASMase plays a crucial role in the internalization of C2 toxin [8]. It has been reported that extracellular proteolytic cleavage regulates ASMase activity [13]. We examined the relationship between lysosomal proteases and ASMase. In the absence of an inhibitor, ASMase activity in the culture medium increased during the first 30 min after C2IIa treatment (Figure 2C). In contrast, ASMase activity decreased in the presence of E64 (Figure 2C), suggesting that cysteine proteases are responsible for the activation of extracellular ASMase.

### 2.3. Effects of Cathepsin siRNAs on C2-Toxin-Induced Cytotoxicity

As shown in Figure 2B, C2IIa induced the release of cathepsins B and L. To examine the role of lysosomal cathepsins in C2 toxin internalization, we used siRNA to suppress the expressions of cathepsins B and L. When MDCK cells were transfected with siRNA targeting cathepsins B (Figure 3A) and L (Figure 3B), siRNA treatments reduced both protein expressions, as compared with intact or negative control (NC)-siRNA. We also evaluated the C2 toxin-caused cell rounding activity. As shown in Figure 3, the toxin caused the cytotoxicity of intact and NC-siRNA-treated cells. This toxin-caused cytotoxicity was blocked by cathepsin B-siRNA but not by cathepsin L-siRNA. These data demonstrate that extracellularly released cathepsin B is needed for C2 toxin internalization into target cells.

## 3. Discussion

The mechanism for initiating C2 toxin internalization in host cells is dependent on their ability to facilitate calcium influx in host cells [8]. C2 toxin-induced Ca^2+^ signaling promotes the exocytosis of ASMase from lysosomes [8]. ASMase hydrolyzes sphingomyelin in the outer plasma membrane and generates ceramide-rich microdomains [9,10,11], which promote endocytosis of C2 toxin [8]. In this study, extracellular cathepsin B released from lysosomes activated ASMase, leading to ceramide production in plasma membranes. Our studies demonstrated that cathepsin B plays a role in C2 toxin endocytosis into target cells.

Previous studies [11,12,13,14] have shown that PFT-induced lysosome exocytosis results in the release of lysosomal hydrolases, such as ASMase and cathepsins, which change the local plasma membrane composition to facilitate PFT internalization via endocytosis. In the present study, we examined the role of lysosomal proteases in C2 toxin endocytosis. Cytotoxicity of C2 toxin against target cells was blocked by cysteine protease inhibitor E64 and lysosome exocytosis blocker BEL. Moreover, E64 inhibited C2IIa internalization, and BEL blocked C2IIa-induced lysosomal marker release from lysosomes. We speculated that C2IIa-induced lysosomal cysteine proteases are related to the internalization of C2 toxin. We then found that CIIa caused the release of cathepsins B and L from lysosomes, that BEL inhibited this C2IIa-induced cysteine protease release, and that cathepsin B knockdown decreased the susceptibility of target cells to C2 toxin. The activity of cathepsin B was higher than that of cathepsin L with C2IIa treatment (Figure 1B). The knockdown of cathepsin B suppressed the cytotoxicity of the toxin more than the knockdown of cathepsin L. As the activity of cathepsin B is high, so there is a possibility that the knockdown effect is strong. On the other hand, since cathepsin L is low activity, it is considered that knockdown of cathepsin L has no effect on the toxin-induced cytotoxicity. On the basis of these findings, our results indicate that cathepsin B released from C2 toxin-treated cells through exocytosis of lysosomes plays an important role in the initial process of internalization of C2 toxin to target cells.

In this study, we did not detect cathepsin D activity in the culture supernatants of C2IIa-exposed cells. Our measurement of lysosomal enzymes secretion indicated that active cathepsin D is not detected in the medium within 30 min, when compared to cathepsin B and L. After synthesis in the endoplasmic reticulum and removal of the signal peptide, procathepsin D is targeted to lysosomes, where further proteolytic cleavage steps produced the mature active enzyme [15,16]. On the other hand, ceramide, which is the product of sphingomyelin hydrolysis by ASMase, can activate cathepsin D [17,18]. Previously, we observed that C2IIa induces ceramide production by ASMase in MDCK cells in a time-dependent manner, with maximum production at 45–60 min [8]. In this experiment, we measured the cathepsin D activity within 30 min after C2IIa exposure. We think that the absence of detectable cathepsin D activity in the medium may have resulted, at least in part, from poor activation of cathepsin D by ASMase-generated ceramide. However, we cannot rule out other potential explanations for the absence of cathepsin D activity, such as activation by other factors [19], the known association of cathepsin D to intracellular membranes [20], and/or extracellular proteolytic processing of the pro-enzyme [15].

It has previously been reported that cathepsins released from exocytosis of lysosomes may participate in the activation of extracellularly released ASMase [13]. In this study, cysteine protease inhibitor E64 inhibited ASMase activity. We think that cathepsin B may be involved in the activation of ASMase in extracellular environments. On the other hand, cathepsin B cleaved the extracellular surface proteins of plasma membranes. Removal of membrane-associated proteins by the protease promoted access of ASMase to membrane substrate sphingomyelin [13,14]. Our results from this experiment also suggest that C2IIa-induced release of lysosomal cathepsin B may be involved in the activation of ASMase and accessibility of ASMase to membrane substrates. In addition, we think that lysosomal cathepsin L may digest cell surface proteins and contribute to membrane access of ASMase. Although much remains unknown about the role of lysosomal cathepsin B released from C2 toxin-treated cells in the endocytic process, the results expand our understanding of the molecular mechanism of C2 toxin internalization to target cells.

## 4. Conclusions

In conclusion, we show evidence for a special role of cathepsin B in the endocytic process of C2 toxin in target cells. C2 toxin induces exocytosis of cathepsin B and ASMase from lysosomes. We think that cathepsin B activates ASMase and increases the accessibility of ASMase to plasma membranes via cell surface protein cleavage by cathepsin B. Ceramide-containing microdomains then invaginate the cell, creating endosomes that facilitate C2 toxin internalization into the cytoplasm.

## 5. Materials and Methods

### 5.1. Materials

C2I and C2II were obtained as described previously [6]. A rabbit antibody against C2II was obtained as described previously [6]. Pepstatin-A, E64, 4-(2-aminoethyl)-benzenesulfonyl fluoride hydrochloride (AEBSF), bromoenol lactone (BEL), *p*-nitrophenyl *N*-acetyl-*β*-*D*-glucosaminide, and protease inhibitor cocktail were obtained from Sigma-Aldrich (Tokyo, Japan). Mouse anti-β-actin antibody and anti-cathepsin L (D-5) antibody were obtained from Santa Cruz Biotehnol. (Santa Cruz, CA, USA). Hanks’ balanced salt solution (HBSS), an Amplex Red Sphingomyelinase Assay Kit, Alexa Fluor 568-conjugated goat antirabbit IgG, Dulbecco’s modified Eagle’s medium (DMEM), Alexa Fluor 488-phalloidin, and 4′,6′-diamino-2-phenylindole (DAPI) were obtained from Thermo Fisher Scientific (Tokyo, Japan). ECL Western blotting detection reagents, horseradish peroxidase-labeled antirabbit IgG, and horseradish peroxidase-labeled anti-mouse IgG were purchased from GE Healthcare (Tokyo, Japan). An anti-cathepsin B (D1C7Y) antibody was purchased from Cell Signaling (Tokyo, Japan).

### 5.2. Cell Cultures and Cytotoxicity Assays

MDCK cells were provided by RIKEN BRC (Ibaraki, Japan). The cells were cultured in culture dishes in 5% CO_2_ at 37 °C in DMEM containing 10% fetal calf serum (FCS), 2 mM glutamine, streptomycin (100 µg/mL) and penicillin (100 U/mL) (FCS-DMEM). In cytotoxicity assays, cells were grown on 48-well tissue culture plates in FCS-DMEM and incubated with the various inhibitors or toxin. Cell morphologies were observed 4 h after toxin treatment, as described previously [8].

### 5.3. β-hexosaminidase Assays

After applications of C2 toxin to MDCK cells, β-hexosaminidase activity in supernatant fluids was analyzed by incubation with 1 mM *p*-nitrophenyl *N*-acetyl-*β*-*D*-glucosaminide as substrate in 0.1 M citrate buffer (pH 4.5) for 1 h at 37 °C [8]. Reactions were terminated by 0.1 M sodium carbonate buffer (pH 9.8). Absorbance values were measured at 405 nm. Enzymatic activities are represented as a percentage of the total enzymatic activity detected in the culture medium and whole cell lysate prepared with 2% Triton X-100.

### 5.4. Determination of Acidic Sphingomyelinase Activities

After applications of C2 toxin to MDCK cells, the supernatant fluids were added to lysis buffer solution (50 mM sodium acetate buffer (pH 5.0), 1 mM EDTA, 1% Triton X-100) containing protease inhibitor cocktail. Activity of ASMase (at pH 5.0) was determined using an Amplex^TM^ Red Sphingomyelinase Assay Kit, as described previously [8].

### 5.5. Small Interfering RNA Silencing and Western Blots

Small interfering RNAs (siRNAs) for cathepsin B, cathepsin D, and negative control siRNA were purchased from QIAGEN (Tokyo, Japan). MDCK cells were treated with siRNA (1 × 10^5^ cells plus 250 pmol siRNA) and electroporated utilizing a Transfection system Neon^TM^ (Invitrogen), using the manufacturer’s protocols. The electroporated cells were inoculated in 24-well microtiter plates and cultured in an antibiotic-free culture medium at 37 °C. Cells were utilized for experiments 48 h after electroporation [8]. Western blots for cathepsin B and cathepsin D were performed on cell lysates, resolved using SDS-PAGE, and transferred to a polyvinylidene fluoride (PVDF) membrane. A Western blot analysis using antibodies against cathepsin B, cathepsin D and beta-actin was performed, as previously described [8].

### 5.6. Immunofluorescence Analysis

Immunofluorescence analysis was conducted as previously described [8]. Briefly, MDCK cells were incubated with C2IIa for 30 min at 37 °C. Cells were fixed in 3% paraformaldehyde and permeabilized with 0.1% Triton X-100. The cells were blocked with 4% BSA in PBS. Subsequently, cells were incubated with rabbit anti-C2II antibody for 1 h in 4% BSA in PBS. The cells were washed four times with PBS and treated with Alexa Fluor 568-conjugated anti-rabbit IgG in 4% BSA in PBS for 1 h. Other labellings were accomplished by staining cell nuclei with DAPI and actin with Alexa Fluor 488-phalloidin. The cells were observed utilizing a confocal laser scanning microscope (Nikon A1, Tokyo, Japan).

### 5.7. Statistical Analyses

Statistical analyses were performed using Easy R (EZR) software (Jichi Medical University, Saitama Medical Center) [21]. A one-way analysis of variance (ANOVA) followed by the Tukey test was utilized to assess differences among three or more groups. Significant differences between two groups were assessed by two-tailed Student’s *t*-test. Differences were considered to be significant for values of *p* < 0.01.

## Figures and Tables

**Figure 1 toxins-13-00272-f001:**
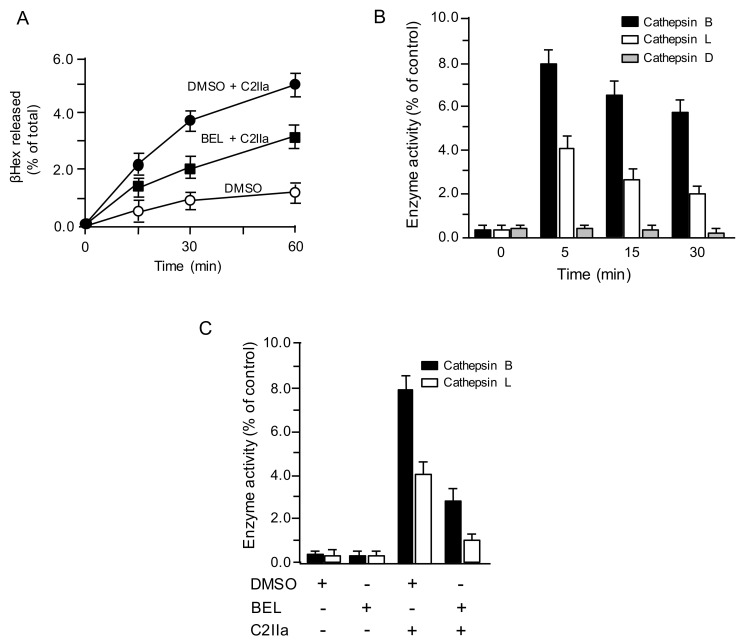
C2 toxin induces lysosomal exocytosis. (**A**) Madin-Darby Canine Kidney (MDCK) cells treated with dimethyl sulfoxide (DMSO) (vehicle) or 25 µM bromoenol lactone (BEL) were incubated with C2IIa (500 ng/mL) for the indicated periods at 37 °C. The culture supernatant fluids were analyzed for β-hexosaminidase (β-Hex) activity. The activities of β-Hex are indicated as the percentage of the total β-Hex activity measured in cells and supernatant. Data are mean values of four independent experiments ± standard deviations. (**B**) MDCK cells were incubated with C2IIa (500 ng/mL) for the indicated periods at 37 °C. (**C**) MDCK cells treated with dimethyl sulfoxide (DMSO) or 25 µM BEL were treated with C2IIa (500 ng/mL) for 30 min at 37 °C. The supernatants were collected and assayed for activities of cathepsin B, cathepsin L and cathepsin D. Data are indicated as percentages of the total enzymatic activity found in the whole cell lysates. Data are mean values of four independent experiments ± standard deviations.

**Figure 2 toxins-13-00272-f002:**
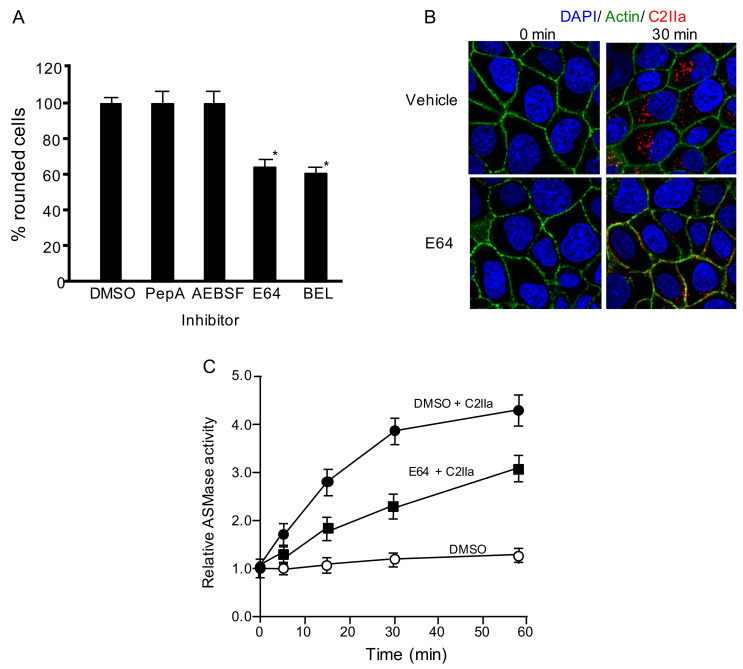
Inhibitors block C2 toxin-caused cytotoxicity in MDCK cells. (**A**) Cells treated with dimethyl sulfoxide (DMSO) (vehicle), 100 µM pepstatin-A (PepA), 100 µM AEBSF, 100 µM E64, or 25 µM bromoenol lactone (BEL) were then treated with C2I (250 ng/mL) and C2IIa (500 ng/mL) for 4 h at 37 °C. Approximately 100 MDCK cells were counted in microscopic photographs, and the percentage of round cells was evaluated. Values from four experiments are given as the mean ± standard deviation (SD). One-way analysis of variance was used to evaluate differences. * *p* < 0.01: significant difference compared to DMSO plus C2-toxin. (**B**) MDCK cells were pretreated with 100 µM E64 for 1 h at 37 °C, and treated with 500 ng/mL C2I and 1000 ng/mL C2IIa at 37 °C. After 30 min, MDCK cells were washed, fixed, and stained with anti-C2IIa antibody, 4,6-diamino-2 phenylindole (DAPI), and Alexa Fluor488-phallodin. C2IIa (red), actin (green) and nucleus (blue) were visualized by confocal microscopy. Representative images of three experiments are shown. Bar: 7.5 µm. (**C**) MDCK cells treated with DMSO or 100 µM E64 were incubated with C2IIa (500 ng/mL) for the indicated periods at 37 °C. Acid sphingomyelinase (ASMase) activity in the culture supernatant fluid was measured, as depicted in Materials and Methods. Intact cells utilized as controls were taken as a baseline level of 1.0. Results are shown as percentages of the values from intact controls. Data are mean values of four independent experiments ± standard deviations.

**Figure 3 toxins-13-00272-f003:**
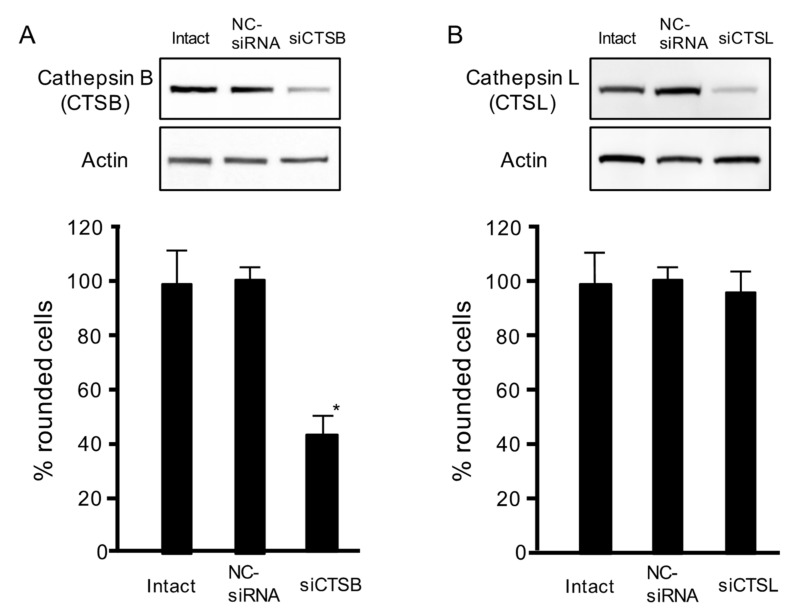
Role of cathepsins B and L on C2 toxin-caused cytotoxicity in MDCK cells. (**A**,**B**) Small interfering RNAs (siRNAs) were utilized to decrease cathepsin B (siCTSB) and L (siCTSL). Nonspecific siRNAs were utilized as a negative control (NC-siRNA). Western blot analysis was used to evaluate the reduced levels of cathepsins B and L. Representative images of three experiments are shown. siRNA-treated cells were treated with C2I (250 ng/mL) and C2IIa (500 ng/mL) for 4 h at 37 °C. Approximately 100 cells were counted in microscopic photographs and the percentage of round cells was evaluated. Values of four experiments are given as the mean ± standard deviation (SD). One-way analysis of variance was used to evaluate differences. * *p* < 0.01: significant difference from NC-siRNA-treated cells plus C2-toxin.

## Data Availability

Not applicable.

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
