# Peer review of "Internalization of Clostridium botulinum C2 Toxin Is Regulated by Cathepsin B Released from Lysosomes"

_toxins, 2021, doi:10.3390/toxins13040272_

Round 1
Reviewer 1 Report
The authors report on experimental evidence that cathepsin B released by MDCK cells by lysosomal exocytosis may be involved in mediating faster uptake of Clostridium botulinum C2 toxin into these cells. The study is based on utilizing inhibitors of endo- and exocytosis as well as of enzyme activities. The experimental design seems adequate to address this question.
However, there are some major points that should be addressed by the authors.
The authors state in the Results section (2.1) that incubation of MDCK cells with C2IIa results in release of βHex to the external medium, and that this could be attenuated by exposure of cells to bromoenol lactone (BEL). BEL is an inhibitor of iPLA2 and negatively affects both endo- and exocytosis. It is also known to inhibit plasma membrane repair processes after cell wounding. Could it be that C2IIa has other effects on MDCK cells than promoting regulated lysosomal exocytosis?
In the same section and in Fig. 1 B the authors state that they find the cathepsins B and L in the culture supernatants of C2IIa-exposed cells, but not cathepsin D. There is no mention in the manuscript on potential reasons. The finding seems strange as cathepsin D definitely is an abundant lysosomal enzyme in MDCK cells.
Fig. 3 shows that treatment of MDCK cells with siRNAs for cathepsin B or C results in a somewhat attenuated expression of these proteases. However, attenuation of cell rounding is only observed in cathepsin B-knockdown cells upon treatment with C2 toxin. This would indicate that only cathepsin B, but not cathepsin L is able to activate ASMase. I find that hard to believe as the substrate specificity of cathepsins is not very high. Has this been shown before or have the authors performed control experiments showing this difference?
A minor point is:
l.3: remove fullstop at the end of the title (no full sentence)
Author Response
The authors report on experimental evidence that cathepsin B released by MDCK cells by lysosomal exocytosis may be involved in mediating faster uptake of Clostridium botulinum C2 toxin into these cells. The study is based on utilizing inhibitors of endo- and exocytosis as well as of enzyme activities. The experimental design seems adequate to address this question.
However, there are some major points that should be addressed by the authors.
The authors state in the Results section (2.1) that incubation of MDCK cells with C2IIa results in release of βHex to the external medium, and that this could be attenuated by exposure of cells to bromoenol lactone (BEL). BEL is an inhibitor of iPLA2 and negatively affects both endo- and exocytosis. It is also known to inhibit plasma membrane repair processes after cell wounding. Could it be that C2IIa has other effects on MDCK cells than promoting regulated lysosomal exocytosis?
Answer: We have reported that C2IIa forms a heptamer on the MDCK cell membrane and induces the influx of Ca into the cytosol (Nagahama et al, Infect. Immun. 85, e00966-16, 2017). It has been clarified that this Ca influx induces lysosomal exocytosis.
In the same section and in Fig. 1 B the authors state that they find the cathepsins B and L in the culture supernatants of C2IIa-exposed cells, but not cathepsin D. There is no mention in the manuscript on potential reasons. The finding seems strange as cathepsin D definitely is an abundant lysosomal enzyme in MDCK cells.
Answer: Our measurement of lysosomal enzymes secretion indicated that active cathepsin D is not detected in the medium within 30 min, when compared to cathepsin B and L. After synthesis in the endoplasmic reticulum and removal of the signal peptide, procathepsin D is targeted to lysosomes, where further proteolytic cleavage steps produced the mature active enzyme (Biochim. Biophys. Acta 1739, 605(2009), Crit. Rev. Oncol. Hematol. 68, 12 (2008)). On the other hand, ceramide, which is the product of sphingomyelin hydrolysis by ASMase, can activate cathepsin D (EMBO J 18, 5252 (1999), Adv. Exp. Med. Biol. 477, 305 (2000)). Previously, we observed that C2IIa induces ceramide production by ASMase in MDCK cells in time-dependent manner, with maximum production at 45 - 60 min (Infect. Immun. 85, e00966-16, 2017). In this experiment, we measured the cathepsin D activity within 30 min after C2IIa exposure. We think that the absence of detectable cathepsin D activity in medium may have resulted, at least in part, from poor activation of cathepsin D by ASMase-generated ceramide. However, we can not role out other potential explanations for the absence of cathepsin D activity, such as activation by other factors (Biochim. Biophys. Acta 1840, 2560 (2014)), the known association of cathepsin D to intracellular membranes (J. Biol. Chem. 263, 6901 (1988)) and/or extracellular proteolytic processing of the pro-enzyme (Crit. Rev. Oncol. Hematol. 68, 12 (2008)).
Fig. 3 shows that treatment of MDCK cells with siRNAs for cathepsin B or C results in a somewhat attenuated expression of these proteases. However, attenuation of cell rounding is only observed in cathepsin B-knockdown cells upon treatment with C2 toxin. This would indicate that only cathepsin B, but not cathepsin L is able to activate ASMase. I find that hard to believe as the substrate specificity of cathepsins is not very high. Has this been shown before or have the authors performed control experiments showing this difference?
Answer: You pointed out an important point. We think as follows. In MDCK cells, the activity of cathepsin B was higher than that of cathepsin L with C2IIa treatment (Fig. 2). Knockdown of cathepsin B suppressed the cytotoxicity of the toxin more than knockdown of cathepsin L. As the activity of cathepsin B is high, so there is a possibility that the knockdown effect is strong. On the other hand, since cathepsin L is low activity, it is considered that knockdown of cathepsin L has no effect on the toxin-induced cytotoxicity.
A minor point is:
l.3: remove fullstop at the end of the title (no full sentence)
Answer: We have removed the period in line 3.
Reviewer 2 Report
In this elegant work, the authors analyze the role of lysosomal proteases in C2 toxin internalization. They find that lysosomal cysteine proteases are required for the endocytosis of C2 toxin, specifically the cathepsin B that act through the activation of ASMase.
Lysosomal cathepsin L was also found to be released by C2IIa, although it was not involved in the endocytosis of C2 toxin; including a comment/hypothesis by the authors on the possible role of cathepsin L would suitable.
This study was carefully executed and the findings are interesting, improve our knowledge on the mechanisms of internalization of the C2 binary toxin in the host cells and deserve publication in Toxins.

Author Response
In this elegant work, the authors analyze the role of lysosomal proteases in C2 toxin internalization. They find that lysosomal cysteine proteases are required for the endocytosis of C2 toxin, specifically the cathepsin B that act through the activation of ASMase.
Lysosomal cathepsin L was also found to be released by C2IIa, although it was not involved in the endocytosis of C2 toxin; including a comment/hypothesis by the authors on the possible role of cathepsin L would suitable.
Answer: We agree with you.We added the below-mentionedexplanations as followedin line166.
In addition, we think that lysosomal cathepsin L may digested cell surface proteins and contribute to membrane access of ASMase.
This study was carefully executed and the findings are interesting, improve our knowledge on the mechanisms of internalization of the C2 binary toxin in the host cells and deserve publication in Toxins.
Answer: Thank you for your recommendation.
Round 2
Reviewer 1 Report
The authors have responded convincingly to my points. However, I would have appreciated if some of their arguments would have been considered for changes in the manuscript as the general reader may raise similar questions that now remain unanswered.
Author Response
The authors have responded convincingly to my points. However, I would have appreciated if some of their arguments would have been considered for changes in the manuscript as the general reader may raise similar questions that now remain unanswered.
Answer. We agree with you. We added the answers to the questions from Reviewer 1 in the manuscript as follows.
1. Could it be that C2IIa has other effects on MDCK cells than promoting regulated lysosomal exocytosis?
Answer We added the explanations as followedin line41-42.
We have reported that C2IIa forms a heptamer on the MDCK cell membrane and induces the influx of Ca2+into the cytosol
2. In the same section and in Fig. 1 B the authors state that they find the cathepsins B and L in the culture supernatants of C2IIa-exposed cells, but not cathepsin D. There is no mention in the manuscript on potential reasons. The finding seems strange as cathepsin D definitely is an abundant lysosomal enzyme in MDCK cells.
Answer We added the explanations as followedin line419-433.
In this study, we did not detected cathepsin D activity in the he culture supernatants of C2IIa-exposed cells. Our measurement of lysosomal enzymes secretion indicated that active cathepsin D is not detected in the medium within 30 min, when compared to cathepsin B and L. After synthesis in the endoplasmic reticulum and removal of the signal peptide, procathepsin D is targeted to lysosomes, where further proteolytic cleavage steps produced the mature active enzyme [15,16]. On the other hand, ceramide, which is the product of sphingomyelin hydrolysis by ASMase, can activate cathepsin D [17,18]. Previously, we observed that C2IIa induces ceramide production by ASMase in MDCK cells in time-dependent manner, with maximum production at 45 - 60 min [8]. In this experiment, we measured the cathepsin D activity within 30 min after C2IIa exposure. We think that the absence of detectable cathepsin D activity in medium may have resulted, at least in part, from poor activation of cathepsin D by ASMase-generated ceramide. However, we can not role out other potential explanations for the absence of cathepsin D activity, such as activation by other factors [19], the known association of cathepsin D to intracellular membranes [20] and/or extracellular proteolytic processing of the pro-enzyme [15].
Added references
15 Benes, P.; Vetvicka, V.; Fusek, M. Cathepsin D--many functions of one aspartic protease. Crit. Rev. Oncol. Hematol. 2008, 68, 12-28.
16 Braulke, T.; Bonifacino, J.S. Sorting of lysosomal proteins. Biochim. Biophys. Acta2009, 1793, 605-614.
17 Heinrich, M.; Wickel, M.; Schneider-Brachert, W.; Sandberg, C.; Gahr, J.; Schwandner, R.; Weber, T.; Saftig, P.; Peters, C.; Brunner, J.; et al. Cathepsin D targeted by acid sphingomyelinase-derived ceramide [published correction appears in EMBO J 2000 Jan 17;19(2):315]. EMBO J. 199918, 5252-5263.
18 Heinrich, M.; Wickel, M.; Winoto-Morbach, S.; Schneider-Brachert, W.; Weber, T.; Brunner, J.; Saftig, P.; Peters, C.; Krönke, M.; Schütze, S.Ceramide as an activator lipid of cathepsin D. Adv. Exp. Med. Biol. 2000, 477, 305-315.
19 Fonović, M.; Turk, B. Cysteine cathepsins and extracellular matrix degradation. Biochim. Biophys. Acta2014, 1840, 2560-2570.
20 Diment, S.; Leech, M.S.; Stahl, P.D. Cathepsin D is membrane-associated in macrophage endosomes. J. Biol. Chem. 1988, 263, 6901-6907.
3. Fig. 3 shows that treatment of MDCK cells with siRNAs for cathepsin B or C results in a somewhat attenuated expression of these proteases. However, attenuation of cell rounding is only observed in cathepsin B-knockdown cells upon treatment with C2 toxin. This would indicate that only cathepsin B, but not cathepsin L is able to activate ASMase. I find that hard to believe as the substrate specificity of cathepsins is not very high. Has this been shown before or have the authors performed control experiments showing this difference?
Answer We added the explanations as followedin line393-415.
The activity of cathepsin B was higher than that of cathepsin L with C2IIa treatment (Fig. 1B). Knockdown of cathepsin B suppressed the cytotoxicity of the toxin more than knockdown of cathepsin L. As the activity of cathepsin B is high, so there is a possibility that the knockdown effect is strong. On the other hand, since cathepsin L is low activity, it is considered that knockdown of cathepsin L has no effect on the toxin-induced cytotoxicity.